# Learning Sparse Gaussian Graphical Models with Overlapping Blocks

**Mohammad Javad Hosseini**[1]    **Su-In Lee**[1,2]

[1]Department of Computer Science & Engineering, University of Washington, Seattle
[2]Department of Genome Sciences, University of Washington, Seattle
{hosseini, suinlee}@cs.washington.edu

## Abstract

We present a novel framework, called GRAB (GRaphical models with overlApping Blocks), to capture densely connected components in a network estimate. GRAB takes as input a data matrix of $p$ variables and $n$ samples and jointly learns both a network of the $p$ variables and densely connected groups of variables (called 'blocks'). GRAB has four major novelties as compared to existing network estimation methods: 1) It does not require blocks to be given *a priori*. 2) Blocks can overlap. 3) It can jointly learn a network structure and overlapping blocks. 4) It solves a joint optimization problem with the block coordinate descent method that is convex in each step. We show that GRAB reveals the underlying network structure substantially better than four state-of-the-art competitors on synthetic data. When applied to cancer gene expression data, GRAB outperforms its competitors in revealing known functional gene sets and potentially novel cancer driver genes.

## 1   Introduction

Many real-world networks contain subsets of variables densely connected to one another, a property called *modularity* (Fig 1A); however, standard network inference methods do not incorporate this property. As an example, biologists are increasingly interested in understanding how thousands of genes interact with each other on the basis of gene expression data that measure expression levels of $p$ genes across $n$ samples. This has stimulated considerable research into the structure estimation of a network from high-dimensional data ($p \gg n$). It is well-known that the network structure corresponds to the non-zero pattern of the inverse covariance matrix, $\mathbf{\Sigma}^{-1}$ [1]. Thus, obtaining a sparse estimate of $\mathbf{\Sigma}^{-1}$ by using $\ell_1$ penalty has been a standard approach to inferring a network, a method called *graphical lasso* [2]. However, applying an $\ell_1$ penalty to each edge fails to reflect the fact that genes involved in similar functions are more likely to be connected with each other and that how genes are organized into functional modules are often not known.

We present a novel structural prior, called *GRAB prior*, which encourages the network estimate to be dense within a *block* (i.e, a subset of variables) and sparse between blocks, where blocks are not given *a priori*. Fig 1B illustrates the effectiveness of the GRAB prior (bottom) in a high-dimensional setting ($p = 200$ and $n = 100$), where it is difficult to reveal the true underlying network by using the graphical lasso (GLasso) (top). The major novelty of GRAB is four-fold:

First, unlike previous work [3, 4, 5], GRAB allows each variable to belong to more than one block, which is an important property of many real-world networks. For example, genes important in disease processes are often involved in multiple functional modules [6], and identifying such genes would be of great scientific interest (Section 4.2). Although existing methods to learn non-overlapping blocks allow edges between different blocks, they use stronger regularization parameters for between-block edges, which decreases the power to detect variables associated with multiple blocks.

Second, GRAB jointly learns the network structure and the assignment of variables into overlapping blocks (Fig 2). Existing methods to incorporate blocks in network learning either use blocks given *a priori* or use a sequential approach to learn blocks and then learn a network given the blocks held fixed. Interestingly, the GRAB algorithm can be viewed as a generalization of the joint learning of the distance metric among $p$ variables and graph-cut clustering of $p$ variables into blocks (Section 3.4)

Third, GRAB solves a joint optimization problem with the block coordinate descent method that is convex in each step. This is a powerful feature that is difficult to be achieved by existing methods to cluster variables into blocks. This property guarantees the convergence of the learning algorithm (Section 3).

Finally, the GRAB framework we presented in this paper uses the Gaussian graphical model as a baseline model. However, the GRAB prior, formulated as $\mathrm{tr}\big(\mathbf{Z}\mathbf{Z}^{\mathsf{T}}|\mathbf{\Theta}|\big)$ (Section 2.2), can be used in any kind of network models such as pairwise Markov random fields.

In the following sections, we show that GRAB outperforms the graphical lasso [2] and existing methods to learn blocks and network estimates [3, 4] on synthetic data and cancer gene expression data. We also demonstrate GRAB's potential to identify novel genes that drive cancer.

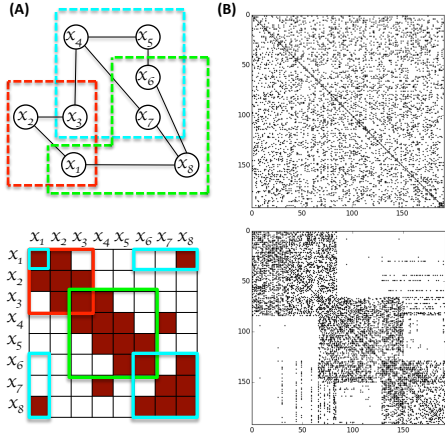

Figure 1: (A) A network with overlapping blocks (top) and its adjacency matrix (bottom). (B) Network estimates of GLasso (top) and GRAB (bottom) in a toy example.

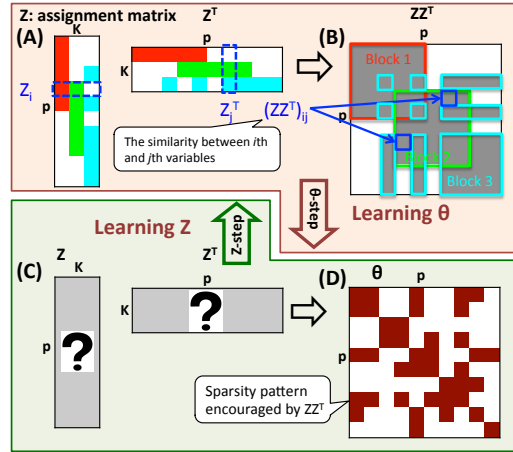

Figure 2: The GRAB framework – an iterative algorithm that jointly learns $\mathbf{\Theta}$ and $\mathbf{Z}$.

## 2 GGM with Overlapping Blocks

### 2.1 Background: High-Dimensional Gaussian Graphical Model (GGM)

We aim to learn a GGM of $p$ variables on the basis of $n$ observations ($p \gg n$). That is, suppose that $\mathbf{X}^{(1)}, \ldots, \mathbf{X}^{(n)}$ are i.i.d. $N(\boldsymbol{\mu}, \boldsymbol{\Sigma})$, where $\boldsymbol{\mu} \in \mathbb{R}^p$ and $\boldsymbol{\Sigma}$ is a $p \times p$ positive definite matrix. It is well known that the sparsity pattern of $\boldsymbol{\Sigma}^{-1}$ determines the conditional independence structure of the $p$ variables; there is an edge between the $i$th and $j$th variables if and only if the $(i, j)$ element of $\boldsymbol{\Sigma}^{-1}$ is non-zero [1]. A number of authors have proposed to estimate $\boldsymbol{\Sigma}^{-1}$ using the *graphical lasso* [2, 7, 8]:

$$\underset{\mathbf{\Theta} \succeq 0}{\mathrm{maximize}} \, \log \det \mathbf{\Theta} - \mathrm{tr}(\mathbf{S}\mathbf{\Theta}) - \lambda \|\mathbf{\Theta}\|_1, \qquad (1)$$

where the solution $\widehat{\mathbf{\Theta}}$ is an estimate of $\boldsymbol{\Sigma}^{-1}$, $\mathbf{S}$ denotes the empirical covariance matrix, and $\lambda$ is a nonnegative tuning parameter that controls the strength of the $\ell_1$ penalty applied to the elements of $\mathbf{\Theta}$. This amounts to maximizing a penalized log-likelihood.

## 2.2 GGM with the Overlapping Block Prior

Here, we present the GRAB prior, formulated as $\text{tr}(\mathbf{Z}\mathbf{Z}^{\mathsf{T}}|\boldsymbol{\Theta}|)$, that encourages $\boldsymbol{\Theta}$ to have overlapping blocks. Let $\mathbf{X} = \{X_1, \ldots, X_p\}$ be variables in the network and $\mathbf{Z}$ be a real matrix of size $p \times K$, where $K$ is the total number of blocks. Each element $-1 \le Z_{ik} \le 1$ can be interpreted as a *score* representing how likely the $i$th variable $X_i$ belongs to the $k$th block $B_k$. The $i$th row of $\mathbf{Z}$, denoted by $Z_i$, can be interpreted as a low-rank *embedding* for the variable $X_i$ showing its block assignment scores. Then, the $(i, j)$ element $(\mathbf{Z}\mathbf{Z}^{\mathsf{T}})_{ij} = \Sigma_{k=1}^{K} Z_{ik} Z_{jk}$ (the dot product of $Z_i$ and $Z_j$) represents the similarity between variables $X_i$ and $X_j$ in their embeddings.

To more clearly understand the impact of the GRAB prior on the sparsity structure of $\boldsymbol{\Theta}$, let us assume a hard assignment model in which we assign variables to blocks. Then, $\mathbf{Z}$ becomes a binary matrix and the sparsity pattern of $\mathbf{Z}\mathbf{Z}^{\mathsf{T}}$ would indicate the region covered by all $K$ blocks (Fig 2A-B). Then, jointly learning $\mathbf{Z}$ and $\boldsymbol{\Theta}$ to increase $\Sigma_{i,j}(\mathbf{Z}\mathbf{Z}^{\mathsf{T}})_{ij}|\Theta_{ij}|$ would encourage $\boldsymbol{\Theta}$ to have a sparsity structure imposed by $(\mathbf{Z}\mathbf{Z}^{\mathsf{T}})$. In the continuous case, it would encourage $|\Theta_{ij}|$ to be non-zero when $X_i$ and $X_j$ have similar embeddings (i.e., a dot product of $Z_i$ and $Z_j$ is large).

Incorporating the GRAB prior into Eq (1) as a structural prior leads to:

$$\underset{\boldsymbol{\Theta} \succeq 0, \mathbf{Z} \in \mathcal{D}}{\text{maximize}} \quad \log \det \boldsymbol{\Theta} - \text{tr}(\mathbf{S}\boldsymbol{\Theta}) - \lambda\Big(\|\boldsymbol{\Theta}\|_1 - \text{tr}\big(\mathbf{Z}\mathbf{Z}^{\mathsf{T}}|\boldsymbol{\Theta}|\big)\Big), \tag{2}$$

where $\lambda$ is a non-negative tuning parameter. We can re-write Eq (2) as:

$$\underset{\boldsymbol{\Theta} \succeq 0, \mathbf{Z} \in \mathcal{D}}{\text{maximize}} \quad \log \det \boldsymbol{\Theta} - \text{tr}(\mathbf{S}\boldsymbol{\Theta}) - \sum_{i,j} \lambda\Big(1 - (\mathbf{Z}\mathbf{Z}^{\mathsf{T}})_{ij}\Big)|\Theta_{ij}|. \tag{3}$$

We use the value of the sparsity tuning parameter $\lambda\big(1 - (\mathbf{Z}\mathbf{Z}^{\mathsf{T}})_{ij}\big)$ for each $(i, j)$ element $\Theta_{ij}$. A network edge that corresponds to two variables with similar embeddings would be penalized less.

The set $\mathcal{D} \subset [-1, 1]^{p \times K}$ contains matrices $\mathbf{Z}$ satisfying the following constraints: (a) $\|Z_i\|_2 \le 1$, where $Z_i$ denotes the $i$th row of $\mathbf{Z}$. This constraint ensures the regularization parameters of all $(i, j)$ pairs of variables are non-negative. (b) $\|\mathbf{Z}\|_F \le \beta$. In addition to the variable specific constraint on each $Z_i$ in (a), we need a global constraint on $\mathbf{Z}$ to prevent all regularization parameters from becoming zero ($\forall i, j : (ZZ^T)_{ij} = 1$). (c) $\|\mathbf{Z}\|_2 \le \tau$, where $\|.\|_2$ of a matrix is its maximum singular value. This constraint prevents the case where all variables are assigned to one block.

There are two hyperparameters, $\beta$ and $\tau$; however we describe below that we set $\tau = \frac{\beta}{\sqrt{K}}$ and that has an effect to guarantee that there are at least $K$ non-empty blocks. In our experiments, we set the hyper-parameter $\beta = \sqrt{\frac{p}{2}}$, which, intuitively, would allow each variable to get on average half of its largest possible squared norm. Given that $\|\mathbf{Z}\|_F^2 = \sum_{i=1}^{p} \sigma_i^2$ where $\sigma_i$ is the $i$th singular value of $\mathbf{Z}$, from the constraint (b), $\sum_{i=1}^{p} \sigma_i^2 \le \beta^2$. We set $\tau = \frac{\beta}{\sqrt{K}}$, where $\tau$ means the upper bound of the maximum singular value, given the constraint (c). This means that there would be at least $K$ non-empty blocks given that the constraint (b) is tight. We show in Section 3 that this choice of hyperparameters makes our learning algorithm simpler (see Lemma 3.2).

## 2.3 Probabilistic Interpretation

The joint distribution over $\mathbf{X}$, $\boldsymbol{\Theta}$ and $\mathbf{Z}$ is as: $P(\mathbf{X}, \boldsymbol{\Theta}, \mathbf{Z}) = P(\mathbf{X}|\boldsymbol{\Theta})P(\boldsymbol{\Theta}|\mathbf{Z})P(\mathbf{Z})$. The first two terms, $\log \det(\Theta) - \text{trace}(S\Theta)$, in Eq (3) correspond to $\log P(\mathbf{X}|\boldsymbol{\Theta})$, the log-likelihood of GGM given a particular parameter $\boldsymbol{\Theta}$ (i.e., an estimate of $\boldsymbol{\Sigma}^{-1}$), as described in Section 2.1. For $\boldsymbol{\Theta} \succeq 0$, $P(\boldsymbol{\Theta}|\mathbf{Z}) = \prod P(\Theta_{ij}|\mathbf{Z})$, where $P(\Theta_{ij}|\mathbf{Z})$ represents a conditional probability over $\Theta_{ij}$ given the block assignment scores of $X_i$ and $X_j$. We use the *Laplacian prior* with the sparsity parameter value $\lambda(1 - (\mathbf{Z}\mathbf{Z}^{\mathsf{T}})_{ij})$. For $\boldsymbol{\Theta} \succeq 0$, $P(\boldsymbol{\Theta}|\mathbf{Z})$ is: $\frac{1}{D}\prod_{(i,j)} \exp(-(\lambda(1 - (\mathbf{Z}\mathbf{Z}^{\mathsf{T}})_{ij}))|\Theta_{ij}|)$, where D is the normalization constant. The prior probability $P(\mathbf{Z})$ is proportional to $D$.

## 2.4 Related Work

To our knowledge, GRAB is the first attempt to *jointly learn* the *overlapping* blocks and the structure of a conditional dependence network such as a GGM. Related work consists of 3 categories:

1) Learning blocks with a network held fixed: This category includes (a) stochastic block model (SBM) [9], (b) spectral clustering [10], and (c) a screening rule to identify non-overlapping blocks based on the empirical covariance matrix [11].

2) Learning a network with blocks given *a priori* and held fixed: This category includes a) a method to solve graphical lasso with group $\ell_1$ penalty to encourage group sparsity of edges within pairs of blocks [12], and b) an efficient learning algorithm for GGMs given a set of overlapping blocks [13].

3) Learning non-overlapping blocks first and then the network given the blocks: (a) Marlin et al. (2009) extend the prior work [12] to identify non-overlapping blocks which are then used to learn a network [3]. (b) Another method assigns each variable to one block, and use different regularization parameters for within-block and between-block edges [14]. (c) Tan et al. (2015) propose to use hierarchical clustering (complete-linkage and average-linkage) to cluster variables into non-overlapping blocks, and apply graphical lasso to each block [4].

## 3 GRAB Learning Algorithm

### 3.1 Overview

Our learning algorithm jointly learns the block assignment scores $\mathbf{Z}$ and the network estimate $\boldsymbol{\Theta}$ by solving Eq (2). We adopt the block coordinate descent (BCD) method to iteratively learn $\mathbf{Z}$ and $\boldsymbol{\Theta}$. Our learning algorithm essentially performs adaptive distance (similarity) metric learning and clustering of variables into blocks simultaneously (Section 3.4). Given the current assignment of variables into blocks, $\mathbf{Z}$, we learn a network among variables, $\boldsymbol{\Theta}$. Then, $|\boldsymbol{\Theta}|$ is used as a similarity matrix among variables to update the assignment of variables to blocks, $\mathbf{Z}$. We iterate until convergence.

Convergence is theoretically guaranteed. Since our objective function is continuous on a compact level set, based on Theorem 4.1 in [15], the solution sequence of our method is defined and bounded. Every coordinate block found by the $\boldsymbol{\Theta}$-step and $\mathbf{Z}$-step is a stationary point of GRAB. We indeed observed the value of the objective function monotonically increases until convergence.

In the following, we show that the BCD method will be convex in each step. We first re-write Eq (2) with all the constraints explicitly:

$$\underset{\boldsymbol{\Theta} \succeq 0, \mathbf{Z}}{\text{maximize}} \quad \log \det \boldsymbol{\Theta} - \text{tr}(\mathbf{S}\boldsymbol{\Theta}) - \lambda \Big( \|\boldsymbol{\Theta}\|_1 - \text{tr}\big(\mathbf{Z}\mathbf{Z}^\mathsf{T}|\boldsymbol{\Theta}|\big) \Big)$$

$$\text{subject to} \quad \|\mathbf{Z}\|_2 \leq \tau, \ \|Z_i\|_2 \leq 1, \ \|\mathbf{Z}\|_F \leq \beta, \ (i \in \{1, \dots p\})). \tag{4}$$

Now, we state the following lemma, the proof of which can be found in the Appendix.

**Lemma 3.1** *Eq (4) is equivalent to the following:*

$$\underset{\boldsymbol{\Theta} \succeq 0, \mathbf{W} \succeq \mathbf{0}}{\text{maximize}} \quad \log \det \boldsymbol{\Theta} - \text{tr}(\mathbf{S}\boldsymbol{\Theta}) - \lambda \Big( \|\boldsymbol{\Theta}\|_1 - \text{tr}\big(\mathbf{W}|\boldsymbol{\Theta}|\big) \Big)$$

$$\text{subject to} \quad rank(\mathbf{W}) \leq K, \ \mathbf{W} \preceq \tau^2 I, \ \text{diag}(\mathbf{W}) \leq 1, \ \text{tr}(\mathbf{W}) \leq \beta^2, \tag{5}$$

*where $\mathbf{W}$ is a $p \times p$ matrix, $K$ means the number of blocks, and $I$ is the identity matrix of size $p$.*[1]

**Corollary 3.1.1** *Suppose that $(\boldsymbol{\Theta}^*, \mathbf{W}^*)$ is the optimal solution of the optimization problem (5). Then, $\boldsymbol{\Theta}^*, \mathbf{Z}^* = U\sqrt{D}$ is the optimal solution of problem 4, where $U \in \mathbb{R}^{p \times K}$ is a matrix with columns containing $K$ eigenvectors of $\mathbf{W}$ corresponding to the largest eigenvalues and $D$ is a diagonal matrix of the corresponding eigenvalues.*

### 3.2 Learning $\boldsymbol{\Theta}$ ($\boldsymbol{\Theta}$-step)

To estimate $\boldsymbol{\Theta}$ given $\mathbf{Z}$, based on Eq (3), we solve the following problem:

$$\underset{\boldsymbol{\Theta} \succeq 0}{\text{maximize}} \quad \log \det \boldsymbol{\Theta} - \text{tr}(\mathbf{S}\boldsymbol{\Theta}) - \textstyle\sum_{(i,j)} \Lambda_{ij}|\Theta_{ij}|, \tag{6}$$

where $\Lambda_{ij} = \lambda(1 - (\mathbf{Z}\mathbf{Z}^\mathsf{T})_{ij})$. This is the graphical lasso with edge-specific regularization parameters $\Lambda_{ij}$. Eq (6) is a convex problem and we solve it by adopting a standard solver for graphical lasso [16].

### 3.3 Learning Z (Z-step)

Here we describe how to learn $\mathbf{Z}$ given $\Theta$. Instead of solving (4), we solve (5) because (5) is a convex optimization problem with respect to $\mathbf{W}$. Interestingly, we can remove the rank constraint, $rank(\mathbf{W}) \leq K$; in Lemma 3.2, we show that with the choice of $\tau = \frac{\beta}{\sqrt{K}}$, the rank constraint is automatically satisfied. This leads to the following optimization problem:

$$\underset{\mathbf{W} \succeq \mathbf{0}}{\text{maximize}} \ \ \text{tr}\big(\mathbf{W}|\Theta|\big)$$
$$\text{subject to} \ \ \mathbf{W} \preceq \tau^2 I, \ \text{diag}(\mathbf{W}) \leq \mathbf{1}, \ \text{tr}(\mathbf{W}) \leq \beta^2. \tag{7}$$

This $\mathbf{W}$-step is a semi-definite programming problem. We solve the dual of Eq (7) that leads to an efficient optimization problem.[2] We introduce three dual variables: 1) a matrix $Y \succeq 0$ for the $\ell_2$ norm constraint, 2) a vector $v \in \mathbb{R}_+^p$ for the constraints on the diagonal and 3) a scalar $y \geq 0$ for the constraint on trace. The Lagrangian is:

$$\mathcal{L}(\mathbf{W}, Y, v, y) = \text{tr}\big(\mathbf{W}|\Theta|\big) + \text{tr}\big((\tau^2 I - \mathbf{W})Y\big) + y(\beta^2 - \text{tr}(\mathbf{W})) + v^T(\mathbf{1} - \text{diag}(\mathbf{W})). \tag{8}$$

The dual function is as:

$$\underset{\mathbf{W} \succeq 0}{\sup} \ \text{tr}\big(\mathbf{W}|\Theta|\big) + \text{tr}\big((\tau^2 I - \mathbf{W})Y\big) + y(\beta^2 - \text{tr}\big(\mathbf{W})\big) + v^\mathsf{T}(\mathbf{1} - \text{diag}(\mathbf{W}))$$

$$= \underset{\mathbf{W} \succeq 0}{\sup} \ \text{tr}\big(\mathbf{W}(|\Theta| - Y - yI - \text{diag}(v))\big) + \tau^2 \text{tr}(Y) + y\beta + v^\mathsf{T}\mathbf{1} \tag{9}$$

$$= \begin{cases} \tau^2 \text{tr}(Y) + y\beta + v^T\mathbf{1} & \text{if } Y \succeq |\Theta| - yI - \text{diag}(v) \\ +\infty & \text{otherwise} \end{cases}.$$

consequently, we get the following dual problem for Eq (7):

$$\underset{Y,y,v}{\text{minimize}} \ \ \tau^2 \text{tr}(Y) + y\beta^2 + v^\mathsf{T}\mathbf{1}$$
$$\text{subject to} \ \ Y \succeq \big(|\Theta| - yI - \text{diag}(v)\big), \ Y \succeq 0, \ y \geq 0, \ v \geq 0. \tag{10}$$

Eq (10) has a closed form solution in $Y$ and $y$ given that $v$ is fixed. The dual problem boils down to:

$$\underset{v \geq 0}{\text{minimize}} \ g(v) = \underset{v \geq 0}{\text{minimize}} \ \tau^2 \sum_{i=1}^{K} \big(C\big)_{+,i} + v^\mathsf{T}\mathbf{1}, \tag{11}$$

where we have replaced $\frac{\beta^2}{\tau^2}$ with $K$ (because $\tau = \beta/\sqrt{K}$). We define $C = (|\Theta| - \text{diag}(v))$ and assume it has eigenvalues $(\lambda_1, \ldots \lambda_p)$ in descending order and $(C)_{+,i} = \max(0, \lambda_i)$. We solve Eq (11) by projected subgradient descent method where the subgradient direction is:

$$\nabla_v g(v) = -\tau^2 \text{diag}\big(U_C 1_K(D_C)U_C^\mathsf{T}\big) + \mathbf{1}. \tag{12}$$

$D_C$ is the diagonal matrix of eigenvalues in descending order and $U_C$ is the matrix containing orthonormal eigenvectors of $C$ as its columns. We define $\mathbf{1}_K(D_C)$ as a binary vector of size $p$ with $j$th element equal to 1 if and only if $j \leq K$ and $\lambda_j > 0$.

After finding the optimal $v^*$, the optimal solution $\mathbf{W}^*$ can be obtained by:

$$\mathbf{W}^* = \underset{\mathbf{W} \succeq \mathbf{0}}{\text{argmax}} \ \text{tr}\big(\mathbf{W}(|\Theta| - \text{diag}(v^*))\big)$$
$$\text{subject to} \ \ \mathbf{W} \preceq \tau^2 I, \ \ \text{tr}(\mathbf{W}) \leq \beta^2. \tag{13}$$

One can see that the solution of problem (13) is $W^* = \tau^2 U_{C^*} 1_{\beta^2/\tau^2}(D_{C^*})U_{C^*}^\mathsf{T} = \tau^2 U_{C^*} 1_K(D_{C^*})U_{C^*}^\mathsf{T}$, where $C^*$, $U_{C^*}$, $D_{C^*}$ and $1_K(D_{C^*})$ are defined similarly to (12). By definition, $1_K(.)$ is a diagonal matrix with at most $K$ nonzeros elements. Therefore, $W^*$ will have rank at most $K$, which means that we do not need the rank constraint on $\mathbf{W}$. This leads to the following lemma.

**Lemma 3.2** *If we set $\tau = \frac{\beta}{\sqrt{K}}$ in (5), the constraint rank$(\mathbf{W}) \leq K$ will be automatically satisfied.*

Finally, we construct $\mathbf{Z}^* = U\sqrt{D}$ as instructed in corollary 3.1.1. Note that in the intermediate iterations, we do not need to compute $\mathbf{Z}$; we need to construct the matrix $\mathbf{Z}^*$ to find the overlapping blocks after the learning algorithm will converge[3].

### 3.4 A special case: K-way graph cut algorithm

Here, we show that GRAB algorithm generalizes the $K$-way graph cut algorithm in two ways: 1) GRAB allows each variable to be in multiple blocks with soft membership; and 2) GRAB updates a network structure $\boldsymbol{\Theta}$, used as a similarity matrix, in each iteration. The proof is in the Appendix.

**Lemma 3.3** *Say that we use a binary matrix $\mathbf{Z}$ (hard assignment) with the following constraints: a) For all variables $i$, $\|Z_i\|_2 \le 1$, where $Z_i$ denotes the ith row of $\mathbf{Z}$. b) For all blocks $k$, $\|Z^k\|_2 >= 1$, where $Z^k$ denotes the kth column of $\mathbf{Z}$. This means that each variable can belong to only one block (i.e., non-overlapping blocks), and each block has at least one variable. Then GRAB is equivalent to iterating between $K$-way graph-cut on $|\boldsymbol{\Theta}|$ to find $\mathbf{Z}$ and solving graphical lasso problem to find $\boldsymbol{\Theta}$.*

## 4 Experimental Results

We present results on synthetically generated data and real data.

**Comparison.** Three state-of-the-art competitors are considered: UGL1 - unknown group $\ell_1$ regularization [3]; CGL - cluster graphical lasso [4]; and GLasso - standard graphical lasso [2]. CGL has two variants depending on the type of hierarchical clustering used: average linkage clustering (CGL:ALC) and complete linkage clustering (CGL:CLC). Each method selects the regularization parameter using the standard cross-validation (or held out validation) procedure.

CGL and UGL1 have their own ways of selecting the number of blocks $K$ [4, 3]. GRAB selects $K$ based on the validation-set log-likelihood in initialization. We initialize GRAB by constructing the $\mathbf{Z}$ matrix. We first perform spectral clustering on $|\mathbf{S}|$, where $\mathbf{S}$ denotes the empirical covariance matrix, then add overlap by assigning a random subset of variables to clusters with the highest average correlation. Then, we project the $\mathbf{Z}$ matrix into the convex set defined in Section 2.2 and form $\mathbf{W} = \mathbf{Z}\mathbf{Z}^\mathsf{T}$. In the $\mathbf{Z}$-step of the GRAB learning algorithm, we use step size $1/\sqrt{t}$, where $t$ is the iteration number and iterate until the relative change in the objective function is less than $10^{-6}$ (Section 3.3). We use the warm-start technique between the BCD iterations.

Evaluation criteria. In the synthetic data experiments (sectoin 4.1), we evaluate each method based on the learned network with the optimal regularization parameter chosen for each method based on only training-set. For the AML dataset (Section 4.2), we evaluate the learned blocks for varying regularization parameters (x-axis) to better illustrate the difference among the methods in terms of their performances. In all experiments, we standardize the data and show the average results over 10 runs and the standard deviations as error bars.

### 4.1 Synthetic Data Experiments

**Data generation.** We first generate $\kappa$ overlapping blocks forming a chain, a random tree or a lattice. In each case, two neighboring blocks overlap each other by $o$ (the ratio of the variables shared between two overlapping blocks). Then, we randomly generate a true underlying network of $p$ variables with density of $20\%$, and convert it to the precision matrix following the procedure of [17]. We generate 100 training samples and 50 validation samples from the multivariate Gaussian distribution with mean zero and the covariance matrix equal to the inverse of the precision matrix.

We consider a varying number of true blocks $\kappa \in \{9, 25, 49\}$ and overlap ratio $o = .25$. For $\kappa = 25$, we consider $o \in \{.1, .25, .4\}$. We vary the number of variables $p \in \{400, 800\}$ for the lattice-structured blocks. The results on the chain and random tree blocks are similar and so we provide only the results for $p = 400$ for these block structures. For all methods, we considered the regularization parameter $\lambda \in [.02, .4]$ with step size .02.

Results. Fig 3 compares five methods when a regularization parameter was selected for each method based on the 50 validation samples. Each of the four plots correspond to different block structure or number of variables. Each bar group corresponds to a particular $(\kappa, o, \eta)$, in which we computed the modularity measure $\eta$ as (fraction of edges that fall within groups - expected fraction if edges were distributed at random), as was done by [18]. Fig 3A shows how accurately each method recovers the true network. For each method $m$, we compared the learned edges ($\mathcal{E}_{Z,m}$) and that from the underlying network ($\mathcal{E}_Z$). By comparing $\mathcal{E}_{Z,m}$ and $\mathcal{E}_Z$, we can compute the precision and recall

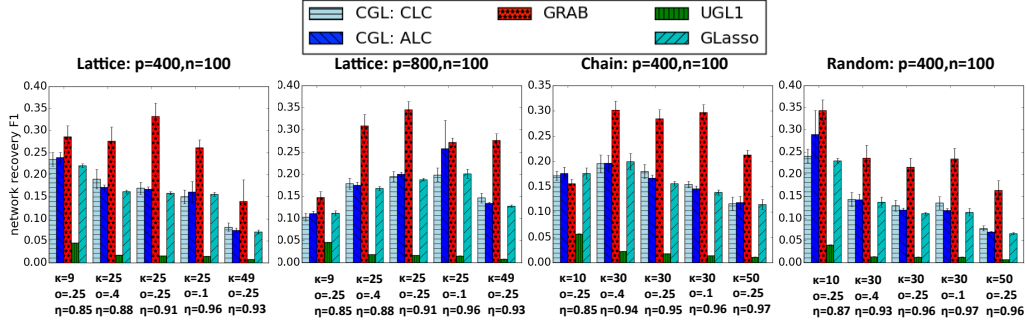

Figure 3: Comparison based on average network recovery $F_1$ on synthetic data from lattice blocks, when $p = 400$ (first panel) and $p = 800$ (second panel), chain blocks (third panel) and random blocks (fourth panel) when $p = 400$. Each bar group corresponds to a particular (number of blocks $\kappa$, overlap ratio $o$, modularity $\eta$).

of network recovery. Since it is not enough to get only high precision or recall, we use the $F_1$ (or F-measure) $= 2 \frac{pr * rec}{pr + rec}$ as an evaluation metric.

A number of authors have shown that identifying the underlying network structure is very challenging in the high-dimensional setting, resulting in low accuracies even on synthetic data [14, 19, 4]. Our results also show that the $F_1$ scores for network are lower than $0.40$. Despite that, GRAB identifies network edges much more accurately than its competitors.

## 4.2 Cancer Gene Expression Data

We consider the MILE data [20] that measure the mRNA expression levels of 16,853 genes in 541 patients with acute myeloid leukemia (AML), an aggressive blood cancer. For a better visualization of the network in limited space (Fig 5), we selected 500 genes[4], consisting of 488 highest varying genes in MILE and 12 genes highly associated with AML: FLT3, NPM1, CEBPA, KIT, N-RAS, MLL, WT1, IDH1/2, TET2, DNMT3A, and ASXL1. These genes are identified by [21] in a large study on 1,185 patients with AML to be significantly mutated in these AML patients. These genes are well-known to have significant role in driving AML.

Here, we evaluate GRAB and the other methods *qualitatively* in terms of how useful each method is for cancer biologists to make discovery from data. For that, we fix the number of blocks to be $K = 10$ across all methods such that we get average of over 50 variables per block, which is considered close to the average number of genes in known pathways [22]. We varied $K$ and obtained similar results.

Genes in the same block are likely to share similar functions. Statistical significance of the overlap between gene clusters (here, blocks) and known functional gene sets have been widely used as an evaluation criteria [23, 5]. We show how to obtain blocks from the learned $\mathbf{Z}$.

**Obtaining blocks from Z.** After the GRAB algorithm converges, we obtain a network estimate $\boldsymbol{\Theta}$ and a block membership matrix $\mathbf{Z}$. We find $K$ overlapping blocks satisfying two constraints: a) maximum number of assignments is $C$; and b) each variable is assigned to $\geq 1$ block. Here, we used $C = 1.3p$. We perform the following greedy procedure: 1) We first run $k$-means clustering algorithm on the $p$ rows of the matrix $Z$.[5] 2) We compute the similarity of variables $i$ to blocks $B_k$ as $\frac{1}{|B_k|} \sum_{j \in B_k} (\mathbf{Z}\mathbf{Z}^\mathsf{T})_{ij}$, where $|B_k|$ is the number of variables in $B_k$. Then, we add overlap by assigning $C - p$ variables to blocks with highest similarity.

To evaluate the blocks, we used 4,722 curated gene sets from the molecular signature database [24] and computed a $p$-value to measure the significance of the overlap between each block and each gene set. We consider the (block, gene set) pairs with false discovery rate (FDR)-corrected $p < 0.05$ to be significantly overlapping pairs. When a block is significantly overlapped with a gene set, we consider the gene set to be revealed by the corresponding block. We compare GRAB with the

methods introduced in section 4.1. Since we only need the blocks for this experiment, we added two more competitors: $k$-means and spectral clustering methods applied to $|\mathbf{S}|$, where $\mathbf{S}$ denotes the empirical covariance matrix. Fig 4 shows the number of gene sets that are revealed by any block (FDR-corrected $p < 0.05$) in each method. GRAB significantly outperforms, which indicates the importance of learning overlapping blocks; GRAB's overlapping blocks reveal known functional organization of genes better than other methods. Fig 4 shows the average results of 10 random initializations.

Fig 5 compares the learned networks $\Theta$ by GLasso (A) and GRAB (B) when the regularization parameters are set such that the networks show a similar level of sparsity. For GRAB, we removed the between-block edges and reordered genes such that the genes in the same blocks tend to appear next to each other. GRAB shows more interpretable network structure, highlighting the genes that belong to multiple blocks.

The key innovation of GRAB is to allow for overlap between blocks. Interestingly, the 12 well-known AML genes are significantly enriched for the genes assigned to 3 or more blocks: FLT3, NPM1, TET2 and DNMT3A belong to 3 blocks while there are only 24 such genes out of 500 genes ($p$-value: 0.001) (Fig 5B). This supports our claim that variables assigned to multiple blocks are likely important. Out of the 24 genes assigned to $\geq 3$ blocks, 12 are known to be involved in myeloid differentiation (the process impaired in AML) or other types of cancer. This can lead to new discovery on the genes that drive AML.

These genes include CCNA1 that has shown to be significantly differentially expressed in AML patients [25]. TSPAN7 is expressed in acute myelocytic leukemia of some patients[6]. Several genes are associated with other types of cancer. For example, CCL20 is associated with pancreatic cancer [26]. ELOVL7 is involved in prostate cancer growth [27]. SCRN1 is a novel marker for prognosis in colorectal cancer [28]. These genes assigned to many blocks and have been implicated in other cancers or leukemias can lead the discovery of novel AML driver genes.

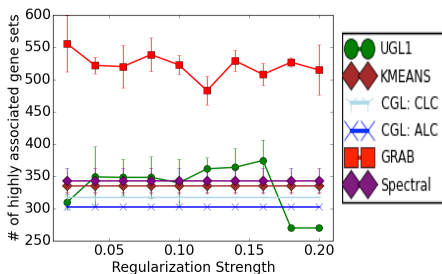

Figure 4: Average number of gene sets highly associated with blocks at a varying regularization parameter. The cross-validation results are consistent with these results.

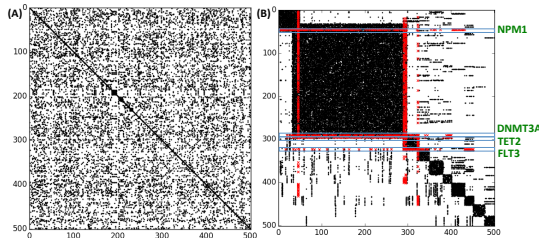

Figure 5: Learned networks of: (A) GLasso and (B) GRAB. For GRAB, we have sorted the genes based on the blocks and highlighted the following 4 genes (out of the 12 highly associated genes with AML) that belong to many blocks: NPM1, FLT3, DNMT3A and TET2.

## 5 Discussion and Future Work

We present a novel general framework, called GRAB, that can explicitly model densely connected network components that can overlap with each other in a graphical model. The novel GRAB structural prior encourages the network estimate to be dense within each block (i.e., a densely connected group of variables) and sparse between the variables in different blocks. The GRAB learning algorithm adopts BCD and is convex in each step. We demonstrate the effectiveness of our framework in synthetic data and cancer gene expression dataset. Our framework is general and can be applied to other kinds of graphical models, such as pairwise Markov random fields.

**Acknowledgements:** We give warm thanks to Reza Eghbali and Amin Jalali for many useful discussions. This work was supported by the National Science Foundation grant DBI-1355899 and the American Cancer Society Research Scholar Award 127332-RSG-15-097-01-TBG.

## Footnotes

[1] In this paper, we assume diag is an operator that maps a vector to a diagonal matrix with the vector as its diagonal, and maps a matrix to a vector containing its diagonal.

[2] The primal problem has a strictly feasible solution $\epsilon I$, where $\epsilon$ is a small number and $I$ is the identity matrix; therefore strong duality holds.

[3] The source code is available at: http://suinlee.cs.washington.edu/software/grab

[4]GRAB runs for 0.5-1.5 hours for 500 genes and up to 20 hours for 2,000 genes on a computer with 2.5 GHz Intel Core i5 processor

[5]This resembles spectral clustering (equivalently, kmeans on eigenvectors of Laplacian matrix)

[6]http://www.genecards.org/cgi-bin/carddisp.pl?gene=TSPAN7

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
