[Supplementary Material · Appendix.pdf]

# 6 Appendix

## 6.1 Proof of Lemma 3.1

***Sketch of proof:*** We use change of variable $\mathbf{W} = \mathbf{ZZ^\mathsf{T}}$. The matrix $\mathbf{W}$ can be decomposed as $\mathbf{ZZ^\mathsf{T}}$ where $\mathbf{Z}$ is $p \times K$, if and only if $\mathbf{W} \succeq 0$ and has rank at most $K$. The constraints $\|\mathbf{Z}\|_2 \leq \tau$, $\|Z_i\|_2 \leq 1$ and $\|\mathbf{Z}\|_F \leq \beta$ are equivalent with the constraints $\mathbf{W} \preceq \tau^2 I$, $\text{diag}(\mathbf{W}) \leq 1$ and $\text{tr}(\mathbf{W}) \leq \beta^2$, respectively. Also, note that the condition that the regularization parameters of all $(i, j)$ pairs of variables are non-negative is satisfied implicitly. For the diagonal elements, we have the constraint $\text{diag}(\mathbf{W}) \leq 1$ explicitly. Without loss of generality, assume $i < j$. We know $\mathbf{W} \succeq 0$, therefore we achieve: $M = \begin{bmatrix} W_{ii} & W_{ij} \\ W_{ji} & W_{jj} \end{bmatrix} \succeq 0$. Having $W_{ii} \leq 1$ and $W_{jj} \leq 1$, we conclude $W_{ij} = W_{ji} \leq 1$.

## 6.2 Proof of Lemma 3.3

***Sketch of proof:*** In the $\mathbf{Z}$-step, to estimate $\mathbf{Z}$ given $\mathbf{\Theta}$ using the BCD method, we solve the following problem based on Eq (2):

$$\underset{\mathbf{Z} \in \mathcal{D}}{\text{maximize}} \ \text{tr}\big(\mathbf{ZZ^\mathsf{T}}|\mathbf{\Theta}|\big), \tag{14}$$

which is equivalent to:

$$\underset{\mathbf{Z} \in \mathcal{D}}{\text{maximize}} \sum_{i,j} (\mathbf{ZZ^\mathsf{T}})_{ij}|\Theta_{ij}|, \tag{15}$$

where $\mathcal{D}$ is defined based on the two constraints (a) and (b) described above. First, note that by solving Eq (14), we will have $\|Z_i\|_2 = 1$, for all variables $i$. Hence, because of the binary assumption, each row of $Z$ will have exactly one element with value 1 and the other elements will be 0.

Then, Eq (15) leads to:

$$\underset{\mathbf{Z} \in \mathcal{D}}{\text{maximize}} \sum_{(i,j) \in \mathcal{F}_Z} |\Theta_{ij}|,$$

which is equivalent to

$$\underset{\mathbf{Z} \in \mathcal{D}}{\text{minimize}} \sum_{(i,j) \notin \mathcal{F}_Z} |\Theta_{ij}|, \tag{16}$$

where $\mathcal{F}_Z$ is the set of edges within blocks $B_1, \ldots, B_K$: $\mathcal{F}_Z = \cup_{k=1}^K \{(i, j)|i, j \in B_k\}$. None of the blocks $B_k$ for all $k$ would be empty due to the constraint (b). Therefore, Eq (16) is equivalent with the $K$-way graph-cut problem on the similarity matrix $|\mathbf{\Theta}|$.

Therefore, in the special case with constraints (a) and (b), $\mathbf{Z}$-step is $K$-way graph-cut on $|\mathbf{\Theta}|$. The GRAB algorithm can be viewed as an iterative BCD method that 1) uses graph-cut as a clustering method to find $\mathbf{Z}$ based on $|\mathbf{\Theta}|$ as a similarity matrix, and 2) learns a network structure of the GGM by solving graphical lasso problem to find $\mathbf{\Theta}$.