[Reviews · NeurIPS 2016]

Reviewer 1

Summary

The authors provide a gaussian graphical model network learning approach that allows elements to belong to overlapping blocks and learns these assignments. The method is evaluated and compared to alternative approaches both on simulated data and cancer data.

Qualitative Assessment

This paper seems to be a contribution to the literature on gaussian graphical model based network modeling. The paper particularly section 3 was rather technically dense. The evaluations were encouraging though not fully satisfying. Specific comments 1. The simulated data seem to only be generated in a manner consistent with the modeling assumptions (e.g. overlapping membership). It would be informative to see how the model performs when the data is generated more consistent with the assumptions of other methods it is compared to. 2. The results in Figure 4 seem quite striking in how consistent all the other methods were in how many gene sets were significant and relative to that how many more were significant with the authors method. Seeing such a large difference raises the question if this increase is all biologically meaningful or is there some type of artificial inflation driving it. It would be informative to show what are these sets (or at least the top ones) that are found significant by GRAB and not the other methods and also report if there are gene sets found by the other method and not GRAB, and to assess to what extent each are biologically meaningful. 3. The biological network was inferred on only 500 genes which is a small fraction of all genes assayed. How scalable is the method to larger sets of genes or all genes assayed? How sensitive is the network inferred if there are important genes that exists but is not included in the input? 4. The procedure for obtaining blocks from Z on page 7 seems somewhat adhoc and contrasts with the principled model based framework in much of the rest of the manuscript. Is there more of a justification for this procedure? 5. It is stated "It is well-known that the network structure corresponds to the non-zero pattern of the inverse covariance matrix" - Should this be stated under any assumptions?

Confidence in this Review

2-Confident (read it all; understood it all reasonably well)


Reviewer 2

Summary

This paper presents a novel method GRAB that jointly learns the network structure and assignment of variables into potentially overlapping blocks. The authors also proposed a reliable optimization algorithm with the block coordinate descent method. Although presented in the context of Gaussian graphical models, GRAB can be generalized to learning of other types of networks. The proposed method is also illustrated in simulations and a real data application to demonstrate its usefulness in uncovering meaningful block structures. Overall it is an interesting and important work to integrate network estimation and community detection into one unified problem. The optimization is done seamlessly. However, there are also some issues to be addressed, mainly regarding the identifiability of the block memberships.

Qualitative Assessment

(1) In this framework, the entries in Z are within [-1, 1]. In the context of stochastic block model, there is also an assignment matrix Z with entries {0, 1} (see Rohe et al 2011 Spectral clustering and the high-dimensional stochastic blockmodel). How does Z here (in the special binary case) compare with the Z matrix in Rohe et al. 2011? (2) In section 2.2, could you explain how the constraint |Z|_F \le beta prevents all regularization parameters from becoming zero? (3) In section 2.2, the constraint (c) |Z|_2 < tau is to prevent the case where all variables are assigned to one block. Why is it not allowed for all variables to be assigned to one block? It's possible that one wants to analyze the interaction network among genes from the same pathway (the same block). In the case of a chain graph, how many blocks are there? (4) In section 4.2, the authors explain how to obtain blocks from the estimated Z matrix. The assignment of the block membership seems rather exploratory. In view of the problem formulation in (3) with the penalty matrix is lambda * (1 - (ZZ^T)_{ij}), there might be identifiability issue. That is, lambda_1 * (11^T - Z_1 Z_1^T) = lambda_2 * (11^T - Z_2 Z_2^T) for (lambda_1, Z_1) \ne (lambda_2, Z_2). Consider the example of a chain graph. In this case, how should one interpret the matrix Z and the following assignment of block memberships? (5) In Section 3.3, equation (8): diag((W)) --> diag(W). Similar corrections should be made one line above equation (9). When introducing equation (11), lambda_i's are used as eigenvalues, whereas lambda was used previously as the penalty parameter. Maybe a different notation is better. In equation (13), the rightmost parenthesis is missing. Further, do you still need the constraint diag(W) \le 1 in equation (13)? (6) In Section 4.2, some details on how the results are obtained are missing. For example, why and how are the between-block edges removed from further analysis? If this is the case, should similar procedure be done for the competing method? In particular, the comparison in Figure 5 seems to be unfair for the Glasso. It is true that GRAB directly returns the block assignment, however one may also recover the blocks based on the estimated adjacency matrix from Glasso. Have you tried building blocks for the Glasso? In that case, it might be fair to compare the estimated blocks for both methods.

Confidence in this Review

3-Expert (read the paper in detail, know the area, quite certain of my opinion)


Reviewer 3

Summary

The authors present GRAB, a model that jointly learns Gaussian graphical models with overlapping blocks (densely connected group of variables). The novel piece in the paper is that ‘overlapping’ blocks are allowed/inferred. The paper gives sufficient detail to why the GRAB prior works. GRAB prior is basically is a neat add to the regularization used in Graphical Lasso. They validate their model on both real (cancer-based gene expression) and simulated datasets and also compare performance with Graphical Lasso and its variants.

Qualitative Assessment

The paper provides a good read with the technical details, related works explained well. Comments. 1. It is not clear to the reader if ‘K’ is learnt in your model. In all the experiments, ‘K’ seems to be an input. if this is the case, you are not learning the number of blocks rather learning the blocks based on a given K. Is this right? The model would have much more value-add if it learnt K as well especially in real-world datasets where you have no idea what K would be. And your low-rank representation Z is of K-dim so a different K would lead to a different representation and hence different block structures! Have you check model performance with different Ks? 2. Lines 38 and 47 - Give necessary details to understand what the variables are. Or give a detailed caption for Fig 2. Fig 2 C-D is not referred to in the text. Again should not the arrow be reversed in Fig C-D since one needs to fix \theta to infer Z? 3. What is the algorithm complexity and what was the runtime especially for the real-world run? Minor comments 1. Equation 4 - what is ‘P’? 2. Section 4.1 Data generation - K versus \mathcal{K}? 3. Consider proof reading the paper - there are a few sentences with typos (for eg 'the the')

Confidence in this Review

3-Expert (read the paper in detail, know the area, quite certain of my opinion)


Reviewer 4

Summary

This paper presented a method named GRAB (GRaphical models with overlApping Blocks), to capture densely connected components in a network estimate. GRAB takes as input a data matrix of p variables and n samples, and jointly learns both a network among p variables and densely connected groups of variables (called ‘blocks’). GRAB outperforms four state-of-the-art competitors on synthetic data. When applied to cancer gene expression data, GRAB outperforms its competitors in revealing known functional gene sets and potentially novel genes that drive cancer.

Qualitative Assessment

The paper reads well. The authors did a great job in presenting the basic motivation and the intuitions behind various equations. The novelty is incremental though. A few questions: - As claimed by the paper - line 40 " Interestingly, the GRAB algorithm can be viewed as a generalization of the joint learning of the distance metric among p variables and graph-cut clustering of p variables into blocks (Section 3.4)". However, Sec3.4 just showed that GRAB algorithm generalizes the K-way graph cut algorithm. Where is the discussion about metric learning ? - The discussion from line 87 is a bit confusing. What does the following sentence mean ? "We avoid to remove the 95 parameter τ for the clarity of writing." - line 103: why " The prior probability P (Z) is proportional to D " ?

Confidence in this Review

2-Confident (read it all; understood it all reasonably well)